# Vehicle and Vessel Detection on Satellite Imagery: A Comparative Study on Single-Shot Detectors

**Tanguy Ophoff [1],\*** , **Steven Puttemans [2]** , **Vasileios Kalogirou [3]**, **Jean-Philippe Robin [3]** and **Toon Goedemé [1]**

1    KU Leuven, EAVISE - Jan Pieter De Nayerlaan 5, 2860 Sint-Katelijne-Waver, Belgium; toon.goedeme@kuleuven.be

2    Flanders Innovation & Entrepreneurship (VLAIO)—Koning Albert II Laan 35, 1030 Brussel, Belgium; steven.puttemans@vlaio.be

3    European Union Satellite Centre—Apdo de Correos 511, Torrejon de Ardoz, 28850 Madrid, Spain; vasileios.kalogirou@satcen.europa.eu (V.K.); jean-philippe.robin@satcen.europa.eu (J.-P.R.)

\*    Correspondence: tanguy.ophoff@kuleuven.be; Tel.: +32-15-31-69-44

**Abstract:** In this paper, we investigate the feasibility of automatic small object detection, such as vehicles and vessels, in satellite imagery with a spatial resolution between 0.3 and 0.5 m. The main challenges of this task are the small objects, as well as the spread in object sizes, with objects ranging from 5 to a few hundred pixels in length. We first annotated 1500 $km^2$, making sure to have equal amounts of land and water data. On top of this dataset we trained and evaluated four different single-shot object detection networks: YOLOV2, YOLOV3, D-YOLO and YOLT, adjusting the many hyperparameters to achieve maximal accuracy. We performed various experiments to better understand the performance and differences between the models. The best performing model, D-YOLO, reached an average precision of 60% for vehicles and 66% for vessels and can process an image of around 1 Gpx in 14 s. We conclude that these models, if properly tuned, can thus indeed be used to help speed up the workflows of satellite data analysts and to create even bigger datasets, making it possible to train even better models in the future.

**Keywords:** satellite; object detection; neural networks; single-shot

## 1. Introduction

Historically, spaceborne remote sensing has been an industry for governments and some heavyweight corporations. However, recent advancements, like the ability to use cost-effective off-the-shelf components (COTS) for cubesats, or the downstream opportunities created by the European Union Copernicus program, have radically changed the industry [1]. This has given rise to new businesses emerging and taking advantage of this geospatial data. To process these huge quantities of data coming from satellites, the space industry needs to speed up and automate workflows, which are traditionally handled by manual operators. Artificial intelligence (AI), being very good at general pattern recognition, lends itself to being a great contender for these tasks.

Imagery intelligence (IMINT) is a discipline which collects information through aerial and satellite means, allowing the monitoring of agricultural crop growth [2], performance of border and maritime surveillance [3,4] and inference of land changes [5] for other applications. Recent advances in computer vision, using deep learning techniques, already allow successful automation of IMINT cases on aerial images [6–8]. Furthermore, locating and segmenting larger objects, e.g., buildings, in satellite imagery is something that is already being used presently [9]. However, detecting smaller objects like vehicles and vessels at these spatial resolutions remains a challenging topic. In this paper, we investigate how

different object detection networks handle the task of vehicle and vessel detection and classification in satellite imagery with a spatial resolution between 0.3 and 0.5 m, where objects appear only a few pixels large in the image.

Traditionally, computer vision algorithms used handcrafted features as an input to a machine learning classifier, in combination with a sliding window approach, in order to perform various tasks like object detection and classification [10–12]. However, with the rise of convolutional neural networks (CNN), deep learning has outperformed these traditional techniques by a significant margin. For the specific case of object detection, there are two different and commonly used approaches: two-stage and single-shot detectors. Two-staged methods like the R-CNN detector [13] will first generate several bounding boxes around potential objects, called region proposals. Each of these proposals will then be run through a classifier to determine whether it actually is an object and what class of object it is. Because they need to run each potential object through a classification algorithm, these techniques are quite slow. Therefore, optimizations have been made in fast R-CNN [14] to share computations between both stages. Faster R-CNN [15] improved upon this even further, by using a deep-learned approach to generate the box proposals, reducing the number of false positive boxes and thus increasing the runtime speeds even further. However, these techniques remain orders of magnitude slower than single-shot detectors. Single-shot detectors [3,4,16–18] are faster because they only process the images through a single neural network, detecting and classifying multiple objects at the same time. Because these detectors work on the entire image at once, they should also be able to use contextual visual information, to detect and classify objects. For the specific case of vessel detection, this means that these single-shot detectors can recognize the structures of sea waves, shorelines, ports, etc. and use that contextual information around the vessels to correctly detect and classify them.

In this paper, we will take a look at the YOLO (You Only Look Once) detector, a well-known and high-performing single-shot detector, and assess its performance for our use case of vehicle and vessel detection in satellite imagery. More specifically, we will compare the YOLOV2 [17] and YOLOV3 [18] detectors, as well as some variations of these, YOLT [3] and D-YOLO [4], which were specifically engineered for aerial object detection in remote sensing applications. We will train these four detectors on our custom dataset for vehicle and vessel detection and will perform various experiments, to assess the strengths and weaknesses of each detector.

In the remainder of this paper, we will first discuss the labeling of ground truth data and how we trained and evaluated the different models (Section 2). Afterwards, in Section 3, we will report various experiments we conducted to assess and understand the performance of the different models. Finally, we will conclude our paper by formulating an answer to the following questions (Section 4):

- Which model is best suited for satellite vehicle and vessel detection?
- What are the different trade-offs between the models?
- Can we consider the problem of automatic satellite object detection solved?

## 2. Materials and Methods

In this section, we will discuss the creation of our dataset (Section 2.1), the different models we used (Section 2.2) and how we trained them (Section 2.3).

### 2.1. Ground Truth Acquisition

To train and assess the performance of our algorithms, we need labeled data. In this paper, we collected images from four different optical satellites (i.e., WorldView-2 and -3, GeoEye and Pleiades), typically used to perform IMINT tasks for security applications. The images were acquired under various acquisition angles varying from 7 to 36 degrees, resulting in different spatial resolutions between 0.3 m and 0.5 m. The images were delivered in 3-band true-color RGB (see Figure 1). We take three images of each type of satellite (further referred to as WV, GE and PL), totaling in nine annotated images, covering around 1500 km$^2$. The entire dataset contains around 53% land and 47% water data.

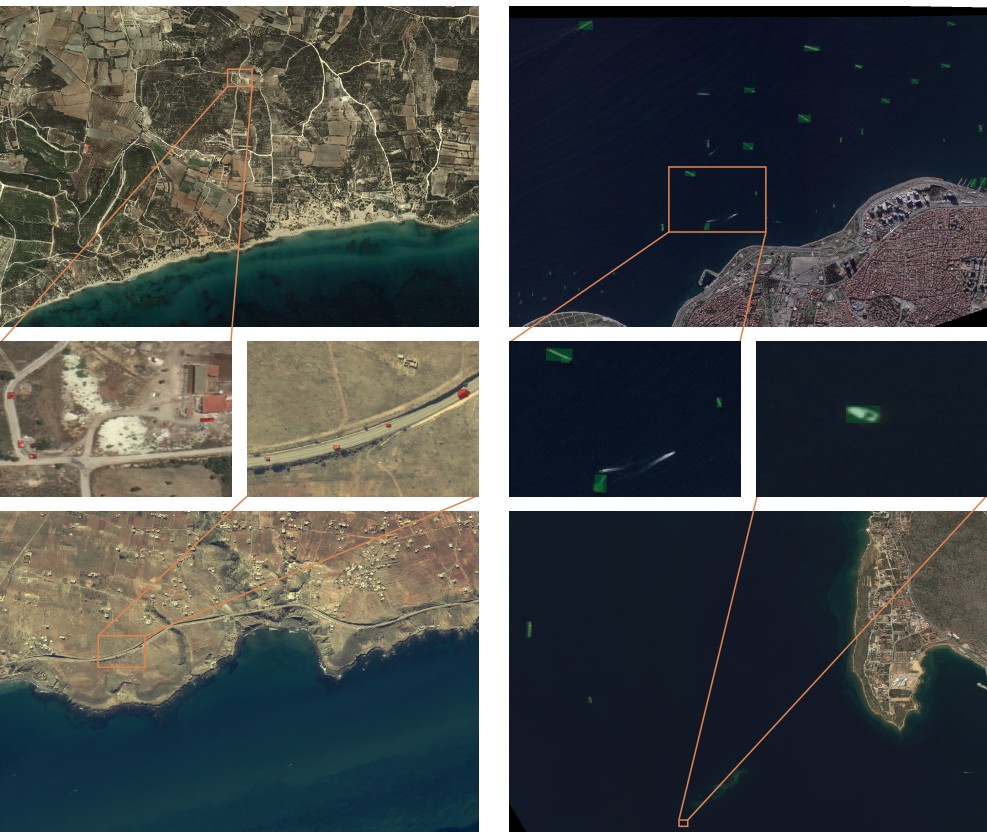

**Figure 1.** Example image crops taken from the training dataset. Vehicles are shown in red, vessels in green.

As not to miss any object during annotation, we decided to split the images in overlapping patches of around $500 \times 500$ pixels. Since annotating the image dataset is a time-consuming and demanding task, we first defined land and water regions in our images. This allows us to only process land region patches during vehicle annotation and water region patches for vessels, speeding up the annotation process. These regions later prove to be beneficial for training and testing our detectors as well. There is little use in training and running a vehicle detector on water bodies and vice-versa for vessel detectors. Doing so will only make it harder for the detector to converge towards an optimal solution and with datasets like the Global Surface Water Explorer [19], which provide water coverage maps with a ground resolution of 30 m, it is possible to automatically create these regions, providing a scene constraint that will increase the performance of a detector.

Figure 2 shows the width and length of the manually annotated vehicles and vessels. This graph shows the two main difficulties detectors will have to cope with, namely that vehicles are only around $10 \times 10$ pixels big and vessels have a huge spread in size, between 5 and almost 500 pixels long. To both train and test the detectors, we split our dataset in a training and testing set, picking two images at random from each type of satellite for training and the remaining image for testing. The number of annotations of each type can be found in Table 1.

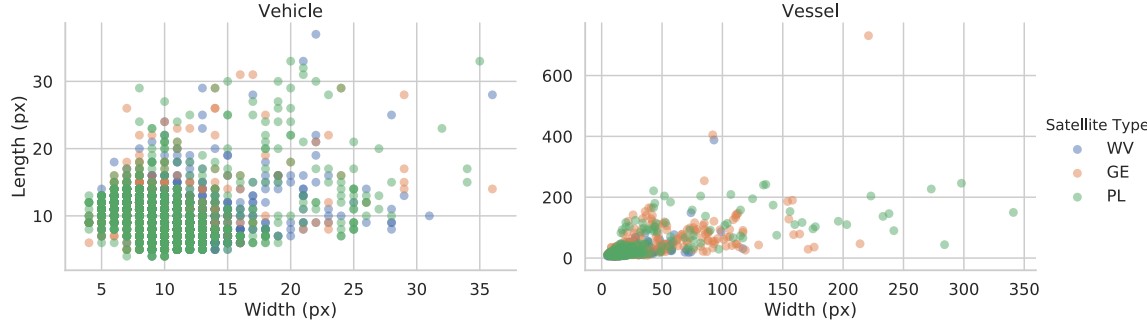

**Figure 2.** Length and Width properties for vehicles and vessels in the annotated dataset.

**Table 1.** Number of annotations in our dataset.

| Satellite | Vehicles | | | Vessels | | |
|---|---|---|---|---|---|---|
| | Train | Test | Total | Train | Test | Total |
| WV | 1477 | 323 | 1800 | 252 | 97 | 349 |
| GE | 413 | 220 | 633 | 301 | 50 | 351 |
| PL | 1318 | 358 | 1676 | 184 | 211 | 395 |
| **Total** | 3208 | 901 | 4109 | 737 | 358 | 1095 |

Detecting vehicles and vessels for operational IMINT purposes can be a useful task, when deployed in large satellite image datasets. However, the value of automation can be considerably higher if the detections can be further classified or labeled on certain categories. The classification scheme adopted in this study, for both vehicles and vessels, considers semantic requirements coming from the IMINT domain (see left columns in Figure 3). For this reason, it includes classes which rarely appear in most vessel detection studies, which focus their research on the detection of larger ships [20]. On the contrary, this study considers also smaller vessels, such as e.g., inflatable rubber boats and skiffs, which can be as small as 3–6 m in length.

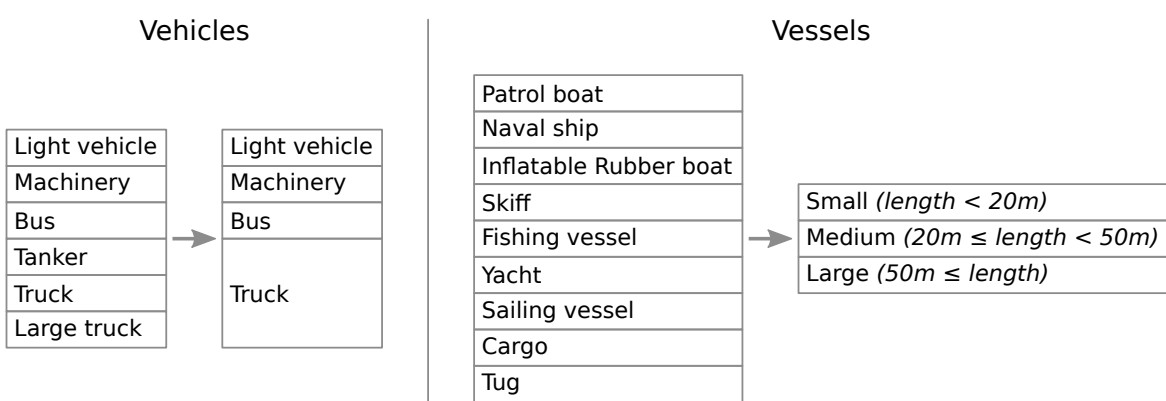

**Figure 3.** Initially defined labels vs. simplified labels.

However, after labeling the vehicles and comparing the labels of two independent researchers, we concluded that the originally defined labels were too specific compared to the spatial resolution of the data. Table 2 shows that there is a huge annotator bias (e.g., when is something considered a truck or a large truck; is any vehicle located in fields to be considered machinery?).

In some cases, due to the semantic or structural similarities existing between the classes (e.g., a sailing vessel vs. a yacht) or due to the limited spatial resolution of some images, the labeling task becomes challenging even for experienced image analysts. In these cases, analysts usually consult

collateral sources of information to assign a label, using Automatic Identification System (AIS) tracking data for vessels or generic contextual information (e.g., port type, or vessel location at port etc.). The scheme was further simplified, to avoid the aforementioned semantic bias and improve the quality of the annotated dataset. Figure 3 shows the original and simplified labels and Table 3 shows the number of objects for each of the simplified labels in our dataset. To decrease the complexity further, we decided to train separate models for the case of vehicle and vessel detection. We will first look at single-class detectors, which are only able to detect vehicles or vessels, but not label them further. Afterwards, we will also train multi-class detectors, which are capable of both detection and classification of our objects.

**Table 2.** Classification differences between two annotators for the original vehicle classes.

| Annotator 2 / Annotator 1 | Light Vehicle | Machinery | Bus | Tanker | Truck | Large Truck |
|---|---|---|---|---|---|---|
| Light vehicle | **1386** | 35 | 9 | 4 | 90 | 25 |
| Machinery | 31 | **7** | 0 | 1 | 3 | 2 |
| Bus | 24 | 1 | **4** | 1 | 0 | 1 |
| Tanker | 4 | 2 | 0 | **0** | 1 | 2 |
| Truck | 179 | 6 | 1 | 0 | **32** | 4 |
| Large truck | 47 | 22 | 0 | 8 | 14 | **18** |

**Table 3.** Number of annotations per label in the dataset.

| Dataset | Light Vehicle | Machinery | Bus | Truck | Small Vessel | Medium Vessel | Large Vessel |
|---|---|---|---|---|---|---|---|
| Train | 2561 | 44 | 106 | 497 | 404 | 258 | 75 |
| Test | 690 | 26 | 16 | 169 | 246 | 93 | 19 |
| **Total** | 3251 | 70 | 122 | 666 | 650 | 351 | 94 |

## 2.2. Network Architectures

In this paper, we compare the YOLOV2 [17], YOLOV3 [18], YOLT [3] and D-YOLO [4] network architectures. All networks were re-implemented with PyTorch [21] and are available in our open-source library Lightnet [22]. This section will provide a quick overview of the different architectures and talk about key differences between them. For a more detailed explanation of each network, we refer to the original research papers.

In 2017 Redmon & Farhadi released YOLOV2 [17], a general-purpose single-shot object detector based on the Darknet19 classification network (see Figure 4a). The network is fully convolutional and concatenates fine-grained features from an earlier feature map in the network to increase the detection accuracy on smaller objects. To be able to combine these feature maps with different spatial resolutions, they invented a reorganization scheme that transforms the higher resolution map into a smaller resolution by dividing it and stacking in the depth dimension.

In 2018 they released an improvement of this network, called YOLOV3 [18]. This network is based on the Darknet53 classification network and is thus much deeper (see Figure 4b). Taking inspiration from feature pyramid networks [23], they concatenate fine-grained features twice, upsampling the lower resolution feature map rather than reorganizing the higher resolution one, and perform predictions from different spatial resolutions as well. These changes resulted in a network that performs better—mostly on smaller objects—but takes a longer time to run.

In 2018, Van Etten released the YOLT model [3], a variation of YOLOV2 that is especially engineered to increase the performance on remote sensing object detection. He noted that YOLOV2 had trouble detecting small objects and thus tried to solve this issue by having less maxpool subsampling operations. Van Etten's network is also slightly smaller (see Figure 4c), to make up for the fact that the reduced maxpool operations increase the number of computations and thus slow down the network.

Lastly, in 2018, Acatay et al. released their model D-YOLO [4], which is a variation of YOLOV2 as well. Trying to solve the same issue as YOLT, they instead opt to take inspiration from feature pyramid

networks [23], with deconvolution operations instead of upsampling (see Figure 4d). This means that all layers except the last 3 operate at the same spatial resolution as YOLOV2, resulting in runtime speeds much closer to it than YOLT.

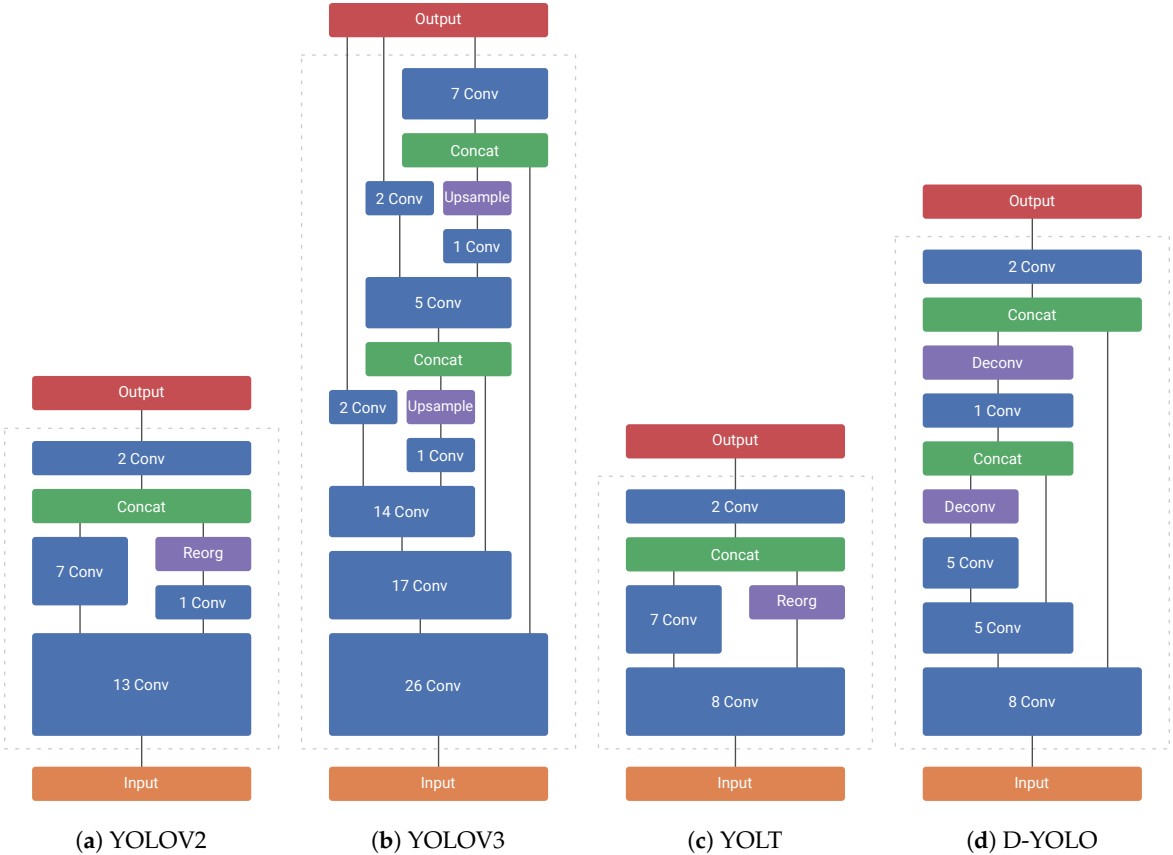

**Figure 4.** Different network architectures.

### 2.3. Training on Satellite Imagery

Satellite images are considerably large (average width-height in our dataset: $34,060 \times 34,877$ pixels) and can thus not be processed by off-the-shelf detectors at once, as the computed feature maps would be too big to fit in GPU memory. Therefore, the images were cut into patches of $416 \times 416$ pixels, with a 10% horizontal and vertical overlap, which get processed individually by the detection networks. Of course, these settings are quite arbitrary and are in fact hyperparameters, which one can tune, depending on the application and the size of the objects that need to be detected. One can then consider the collection of all these patches as the entire training dataset, but filtering the patches and only keeping those which actually contain objects to detect, will increase the performance of the detectors. Indeed, every part of a patch that does not contain an object is already providing a negative example for the detector to train on. This number of negative examples has empirically showed to be enough to successfully train a model and as such we do not need more negative examples by adding these empty patches. However, to correctly evaluate the performance of the different models, we do need to include all patches during the testing phase of the detectors. Because we allow for overlap between the different patches, the non-maximal suppression post-processing step should be used after combining the bounding boxes of the different patches, to remove duplicate detections.

When training a network, one needs to set up some parameters which influence how the model trains. These are called hyperparameters and correctly fine-tuning them makes the difference between a good and a bad model. Moreover, these hyperparameters need to be tweaked for every new use case and can differ wildly, depending on the dataset, model, etc. Hyperparameters like learning rate,

momentum, training time, etc. have a big influence on the training outcome and were thus tweaked manually, by trial and error (combined with the researcher's experience and insights), to try and achieve the best performance. However, we observed that some lesser known hyperparameters that influence the yolo-specific loss function, had a significant impact on the achievable accuracy of the models as well. Indeed, if we break down what single-shot detection networks need to predict, we can find three distinct tasks: detecting objects, finding bounding box coordinates and optionally classifying these objects. These three different tasks each have a different loss function, which get combined by taking the weighted sum of these different sub-losses. The task of detecting objects can be further broken down into two parts, namely how important it is to predict a high confidence when there is an object and how important it is to predict a low confidence when there is no object. When training a model, one can change the weight of each of these four sub-losses, effectively modifying the relative importance of these different parts and thus the final behavior of the model.

Care must be taken not to overfit a model on the testing dataset when selecting these hyperparameters, and to that end we chose to tune our parameters on a subset of our data. We thus trained and evaluated the four different models on the GE subset of our data and tweaked the hyperparameters in order to achieve the highest possible average precision at an IoU threshold of 50% (see Table 4). Once we found these hyperparameters, we used these exact values for the remainder of our experiments.

**Table 4.** Manually tuned hyperparameters for the different models on the GE subset of our data. Please note that not all hyperparameters are shown.

| Hyperparameter | Vehicles | | | | Vessels | | | |
|---|---|---|---|---|---|---|---|---|
| | YOLOV2 | YOLOV3 | YOLT | D-YOLO | YOLOV2 | YOLOV3 | YOLT | D-YOLO |
| Learning rate | 0.001 with division by factor 10 after 4000, 7000, 10,000 batches | | | | | | | |
| Momentum | 0.9 | 0.9 | 0.9 | 0.9 | 0.9 | 0.9 | 0.9 | 0.9 |
| Batch size | 32 | 32 | 32 | 32 | 32 | 32 | 32 | 32 |
| Max batches | 15,000 | 15,000 | 15,000 | 15,000 | 15,000 | 15,000 | 15,000 | 15,000 |
| Object Scale | 5.0 | 7.0 | 6.0 | 10.0 | 5.0 | 5.0 | 5.0 | 5.0 |
| No-Object Scale | 1.0 | 3.0 | 1.5 | 3.0 | 1.0 | 2.0 | 2.5 | 2.0 |
| Coordinate Scale | 2.0 | 1.0 | 2.0 | 1.0 | 1.5 | 1.5 | 1.0 | 1.0 |
| Classification Scale | 1.0 | 1.0 | 1.0 | 1.0 | 1.0 | 1.0 | 1.0 | 1.0 |

## 3. Results

In this section, we will discuss the experiments we ran with our trained networks, to better understand the strengths and weaknesses of each detection model for satellite object detection.

### 3.1. Precision–Recall

The first and most obvious way to compare object detection networks is to use Precision (P) Recall (R) curves and the Average Precision (AP) metric. These metrics have been a standard for object detection for a long time and provide an easy way to compare different architectures, as well as giving an insight into different working points for using these detectors.

Looking at the AP of the four detectors on the entire dataset (see Figure 5), we can see that for the case of satellite object detection, YOLOV2 has a 20–30% lower AP than the other detectors. For the specific case of vessel detection, the YOLT detector seems to have a really interesting performance, beating both D-YOLO and YOLOV3 by over 4% and 8% respectively. These results can also be seen on the qualitative examples in Figure 6.

Vehicle Detection [IoU >= 50%]

YOLT [50.29%]
DYOLO [54.229%]
YOLOv2 [21.803%]
YOLOv3 [54.798%]

Vessel Detection [IoU >= 50%]

YOLT [62.703%]
DYOLO [58.273%]
YOLOv2 [36.727%]
YOLOv3 [54.332%]

**Figure 5.** PR-curves of our detectors for vehicles and vessels. The IoU threshold was set at 50%. AP values are written in the legend.

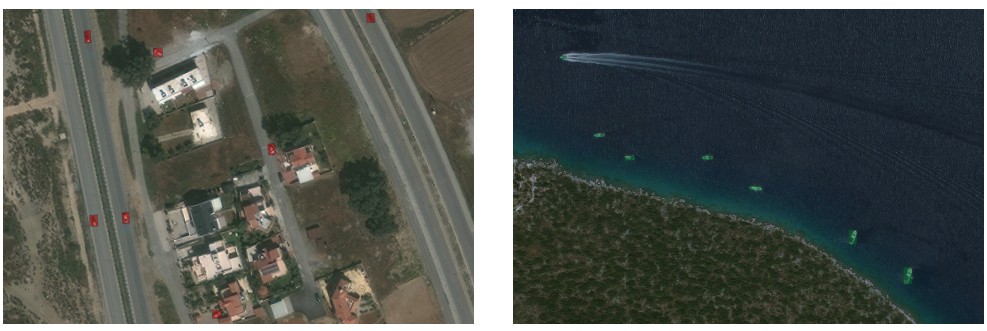

(**a**) Annotation bounding boxes

**Figure 6.** *Cont.*

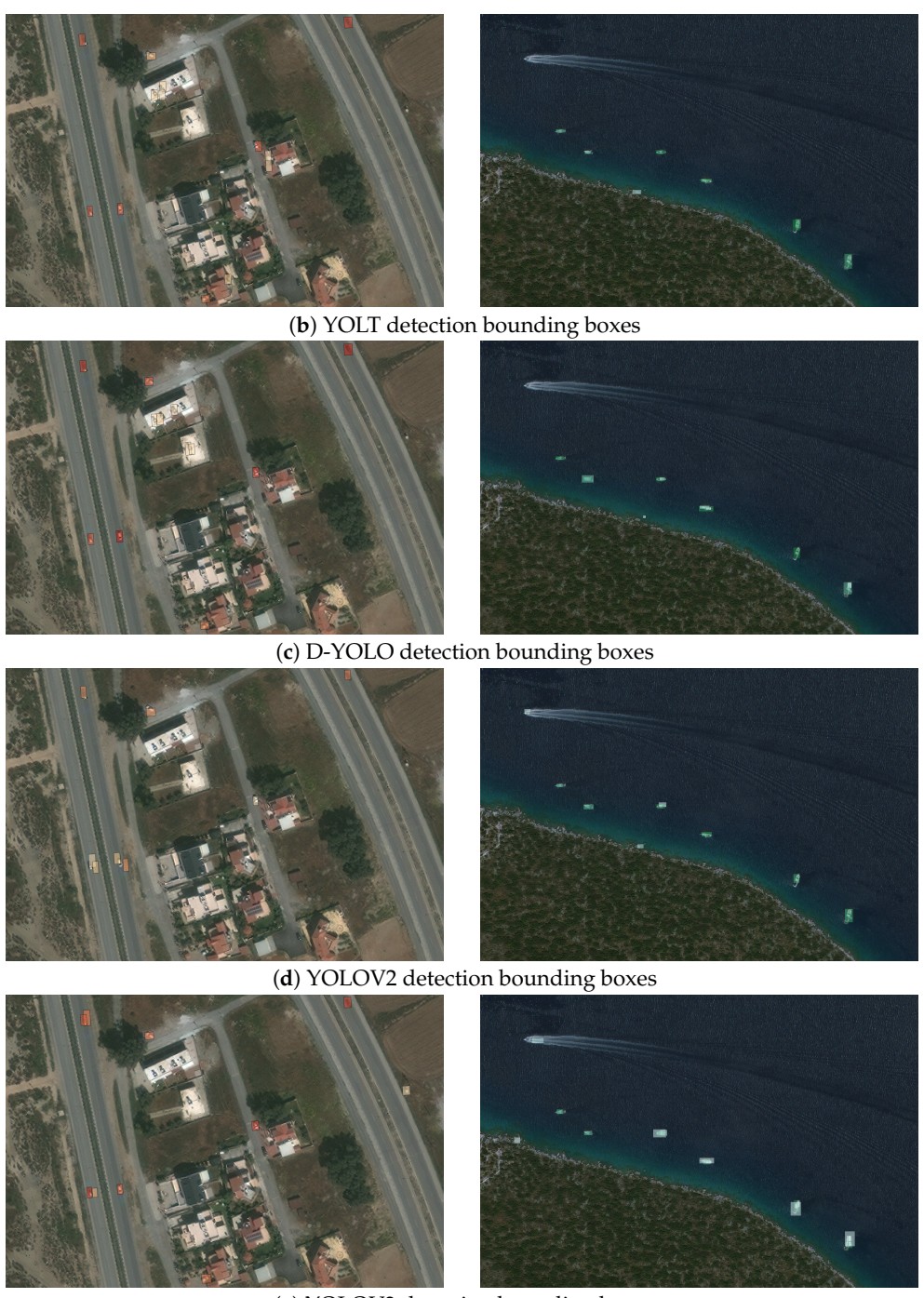

**Figure 6.** Qualitative comparison of our models. Darker colors mean higher confidences. Please note that we only show detections with a confidence higher than 10%. Please note that these images are small crops, taken from the full satellite images, to be able to show the objects that need to be detected. (**a**) Annotation; (**b**) YOLT; (**c**) D-YOLO; (**d**) YOLOV2; (**e**) YOLOV3.

### 3.2. Localization Error

Traditional PR/AP metrics work at an Intersection over Union (IoU) threshold of 50%, meaning they count a detection as a true positive as long as it overlaps with a ground truth object with an IoU of at least 50%. This means that these metrics do not allow you to assess exactly how accurate the bounding boxes from a detector are aligned with the ground truth. One way to be able to compare the localization accuracy of different detectors is to compute the AP of these detectors at different IoU thresholds, which is what we did in Figure 7.

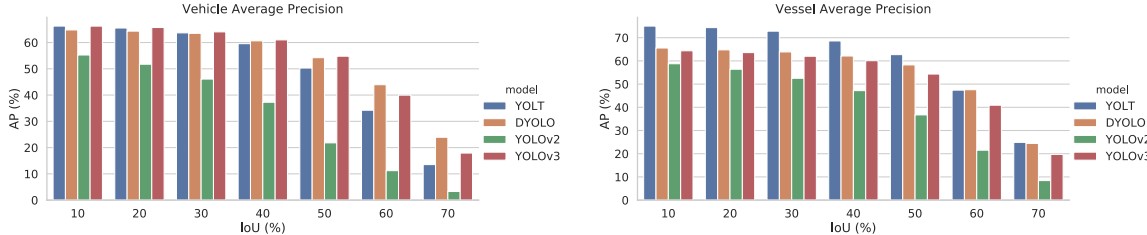

**Figure 7.** Average Precision of the different detectors at increasing IoU thresholds.

These figures show that D-YOLO and YOLOV3 provide the most accurate bounding boxes, as they relatively maintain their AP levels, for increasing values of IoU threshold. While YOLT seems to be on par with D-YOLO and YOLOV3 for vessels, this graph shows that for vehicles, YOLT has difficulties to accurately define bounding boxes. As vehicles are considerably smaller targets, this probably indicates that we are reaching the limits in terms of bounding box localization for YOLT.

It also shows that YOLOV2 performs considerably good at finding objects, but it demonstrates difficulty to accurately delineate them in bounding boxes. Indeed, when comparing the detectors with a lower IoU threshold of 10%, we find that YOLOV2 only has a 10–15% accuracy drop compared to the other detectors and when comparing at a threshold of 60%, this difference increases to 20–35%.

### 3.3. Pretrained Weights

When training neural networks for a certain task like object detection, it is known to be beneficial to start from pretrained weights from a similar network, which might have been trained for a different task. This is done to cut the training times and size of the dataset needed to train a network. In our case we start from weights from a classification network (*i.e.* Darknet19, Darknet53) trained on the ImageNet dataset [24].

However, the difference between the ImageNet dataset and the satellite images used in this study is significant, as the first is based on "natural" real-world images, while the latter depicts the world from a different view—from several kilometers above the Earth. Still, certain studies suggest that the low- and mid-level features extracted from ImageNet-trained CNNs have high potential in other tasks [25].

To evaluate the impact of pretrained weights, we compare training the networks from pretrained ImageNet weights and from pretrained weights originating from the Dataset for Object Detection in Aerial Images (DOTA) [6]. While the objects from DOTA and our dataset are not the same and not of the same size, it is highly likely that the pretrained weights from DOTA should be more attuned to satellite footage. Moreover, DOTA being an object detection dataset, these weights should also be more geared towards detection, compared to the weights from ImageNet, which is a classification dataset. However, only the first few layers can be loaded with pretrained weights from the ImageNet-trained classification networks, as they are the only that remain the same between the classification and detection networks (e.g., until the 23th layer for Darknet19/YOLOV2). To be able to compare our results, only those layers were loaded with pretrained weights from DOTA as well.

Comparing the graphs in Figure 8 with the previous ones (Figures 5 and 7), one can infer that using pretrained weights from a similar domain does offer advantages over weights from a different domain like ImageNet. Furthermore, the D-YOLO architecture seems to be outperforming YOLT in these experiments for both tasks of vehicle (+7.3%) and vessel detection (+4%), indicating that this network is more capable than YOLT, but might need more training data when starting with ImageNet weights.

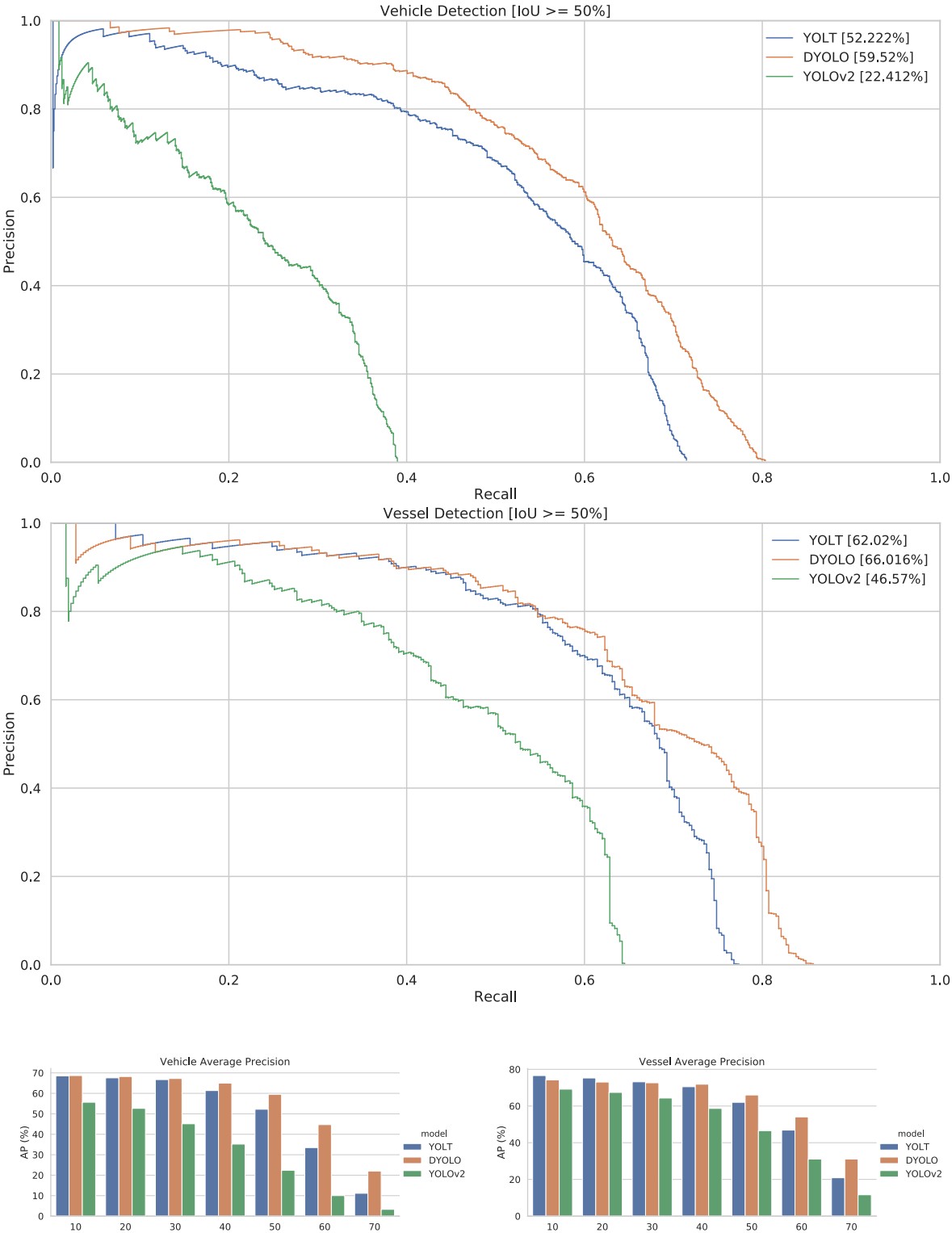

**Figure 8.** PR-curves and AP-IoU graphs of the different detectors trained starting from DOTA pretrained weights. Please note that the YOLOV3 network is not shown, as we could not find pretrained DOTA weights for this network.

### 3.4. Image Variation

Our dataset consists of images from 3 different types of satellite: WorldView (WV), GeoEye (GE) and Pleiades (PL). To test whether the different models can cope with the variation in spatial resolution, we performed an ablation study, training and testing on only subsets of our data containing certain types of satellite imagery. Bear in mind that for these training routines, we did not modify any of the hyperparameters, still using the ones adapted on only the images from the GE subset.

The results suggest that our models can cope with imagery from different sources and varying qualities (see Figures 9 and 10). When comparing the AP of our models when using the entire dataset (last row: WV+GE+PL) with the AP of a single type of satellite (first three rows: WV, GE and PL), we can see a general trend where the performance increases when we use the combined data. This indicates that there is a bigger advantage in using more images, compared to the disadvantage of added complexity, brought by combining images from different types of satellites. Thus, training on our entire dataset results in more robust detection models.

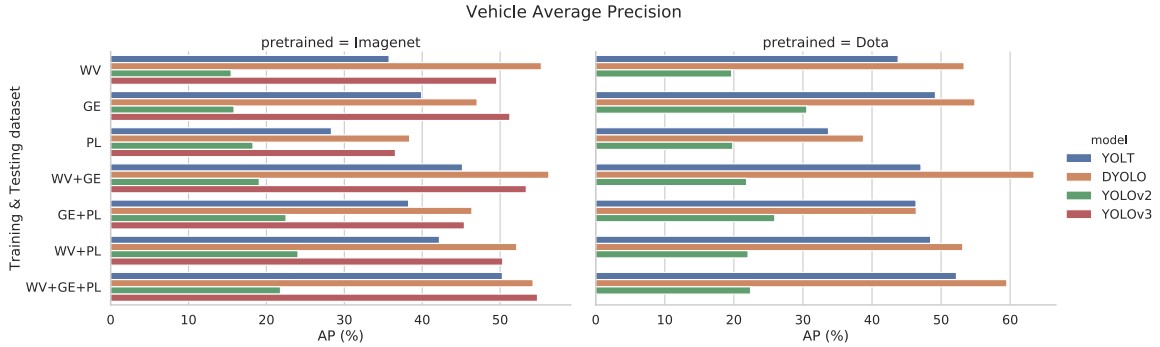

**Figure 9.** Average Precision of the different vehicle detectors for different subsets of our data.

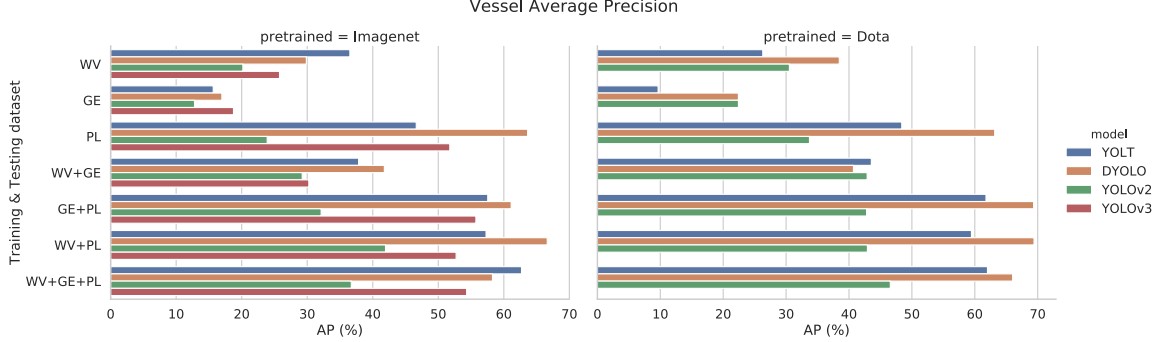

**Figure 10.** Average Precision of the different vessel detectors for different subsets of our data.

### 3.5. Multi-Label Detection

Besides being able to detect vehicles and vessels, we are also interested to know what kind of vehicle or vessel is found at a certain location. However, the problem of classifying vehicles at these resolutions remains a very challenging task even for image analysts, who use additional data or context to successfully label several object categories.

Nevertheless, we wanted to investigate whether our chosen algorithms could differentiate between these different types, solely based on visual information. Because single-shot detectors are also capable of performing classification, the most straight-forward approach was to train a multi-class detector. In this experiment, we chose the best performing detectors on the entire dataset (YOLT for vessels and D-YOLO for vehicles) and fine-tuned their training to perform multi-class detection.

Figure 11 shows the PR-curve of our single-class and multi-class detectors, when disregarding the class labels. These graphs clearly demonstrate that adding a classification task to these networks does not deteriorate their detection capabilities, but even marginally increases their detection performance.

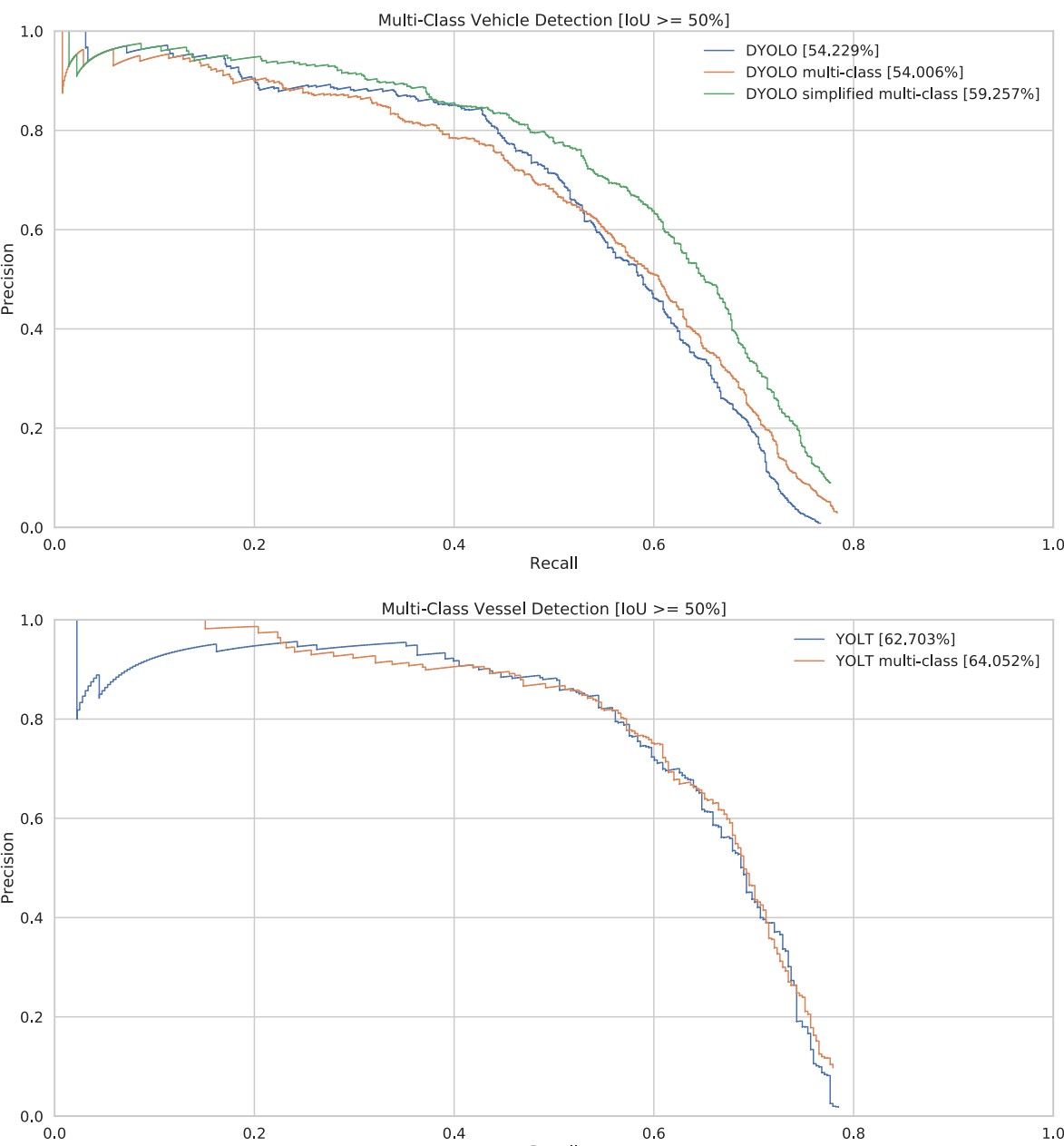

**Figure 11.** PR-curves of our multi-class detectors. AP values are written in the legend. Please note that for these curves, we disregard the labels, to only measure the detection accuracy.

Looking at the top-1 classification accuracy (see Table 5), the results seem promising and useful at a first glance. However, when we analyze the confusion matrices more closely (see Table 6), we can see that for the case of vehicle detection our classification pipeline just labels most of the objects as light vehicles. Our annotation statistics (see Table 3) show that there is a huge class imbalance between the different categories, with light vehicles being by far the biggest category. The detector thus learned to classify most of the objects as light vehicles, as this indeed gives the best results overall. This can also be seen when plotting the individual PR-curves of the different classes (see Figure 12).

**Table 5.** Top-1 accuracy of our multi-class detectors.

| | Vehicle | Vessel |
|---|---|---|
| Top-1 (%) | 58.86 | 68.72 |

**Table 6.** Confusion matrix of the classification results of our multi-class detectors.

| Annotation / Detection | Light Vehicle | Machinery | Bus | Truck | False Negative |
|---|---|---|---|---|---|
| Light vehicle | 452 | 2 | 8 | 77 | 147 |
| Machinery | 4 | 4 | 2 | 3 | 13 |
| Bus | 3 | 0 | 4 | 8 | 1 |
| Truck | 47 | 8 | 11 | 68 | 35 |
| False positive | 14,653 | 3020 | 1428 | 6395 | |

| Annotation / Detection | Small Vessel | Medium Vessel | Large Vessel | False Negative |
|---|---|---|---|---|
| Small vessel | 169 | 12 | 0 | 65 |
| Medium vessel | 17 | 64 | 0 | 12 |
| Large vessel | 0 | 4 | 13 | 2 |
| False positive | 1535 | 703 | 352 | |

The results for multi-class vessel detection are more promising, with most of the misclassification errors made among the small and medium vessels. As the classes were arbitrarily defined at 20 m in length, it is possible that the detector faces difficulties in classifying vessels that are around that length. The individual PR-curves of the different vessel classes are also closer to one another, demonstrating that a more balanced dataset is key to achieving good overall results.

Please note that our confusion matrices contain an extra 'False positive' row and 'False negative' row. This is because we are working with detectors and thus, they indicate erroneous and missing detections, respectively. The confidence threshold of the detectors was set to 0.1%, resulting in a lot of false positives, but we believe this is the best for showing the classification potential. Indeed, setting the threshold higher could result in an object that was correctly classified, to be discarded because its confidence value is not high enough.

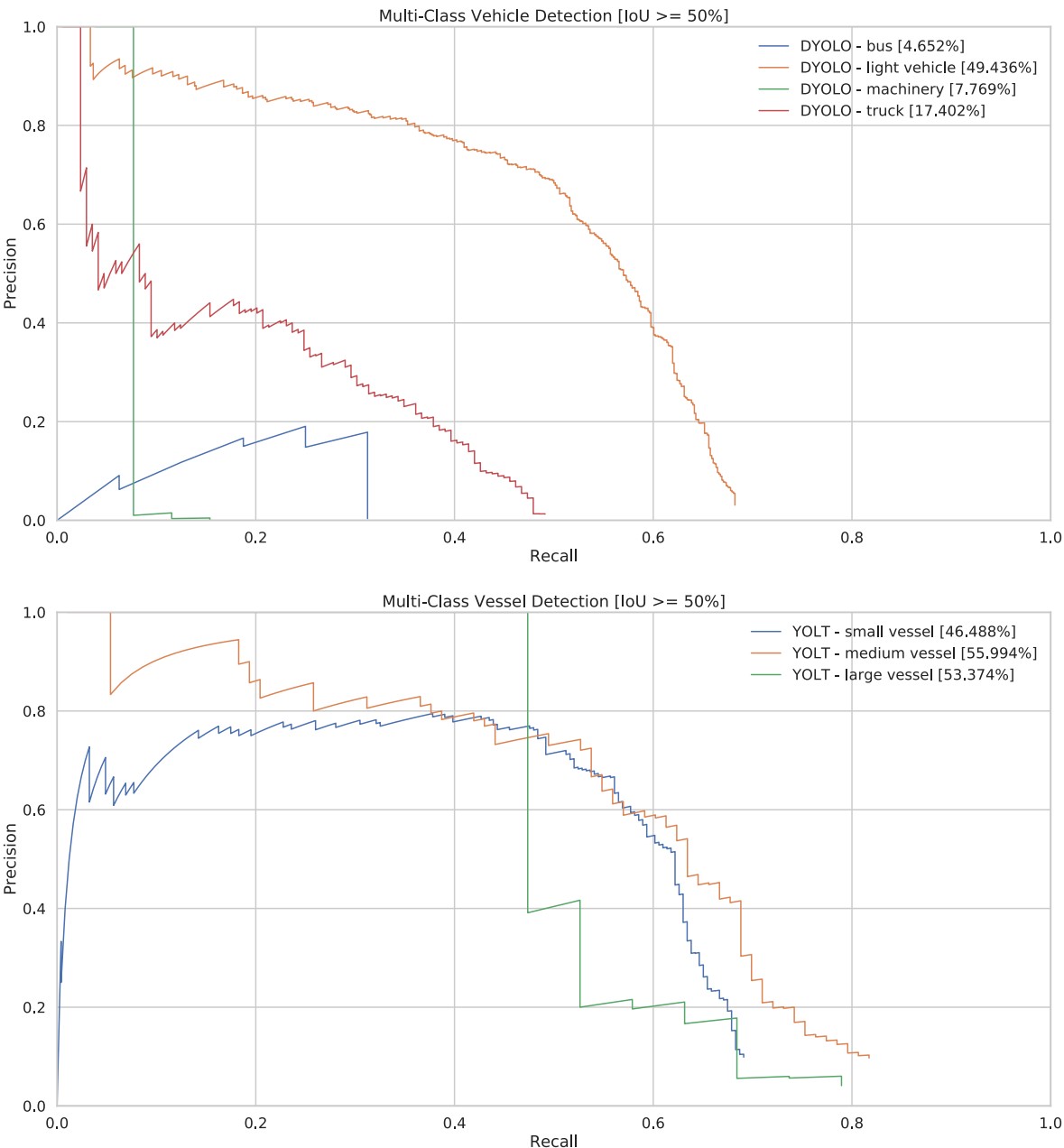

**Figure 12.** PR-curves per label of our multi-class detectors. AP values are written in the legend.

*3.6. Speed*

Besides performance, speed is another important factor which impacts the operational use of the detectors. As the current trend of satellite on-board processing is developing [26], it is important that the AI algorithms are adapted to use few resources and run in a timely fashion on the constrained hardware. Speed is also important when running the detectors in e.g., a datacenter, solely because of the sheer amount of data that needs to be processed.

To measure the speed of a detector, we need to choose a working point at which it will operate and thus we have to select a threshold to filter which bounding boxes get accepted and which do not. This has a significant influence on the post-processing time and as such it is important to set this up correctly. Because we have no specific precision or recall constraints for this study, we use the threshold with the highest F1-score of our precision and recall (see Figure 13).

Figure 14 shows the inference time for both the network and post-processing on a 416 × 416 image patch and was measured by averaging the time of running through the entire test set. The first things to note are that the tests give similar results for both vehicles and vessels and that the post-processing time can be neglected compared to the runtime of the network. When comparing the 4 different networks, we can clearly see that D-YOLO and YOLOV2 are the fastest, with an inference time of about 4 ms per patch. YOLT is slightly slower with 6 ms per patch and finally YOLOV3 is the slowest, taking twice as long to process a patch with 8 ms per patch.

The average timings of a single patch might appear to be insignificant. However, one needs to take into account that typical satellite images as the ones used in this study consist of 3500 patches on average. As a result, the processing time of one image for D-YOLO and YOLOV3 can be around 14 and 28 s respectively, which is significant and can be a determining factor for choosing one model over the other, seeing as the AP is similar for the different models (see Figure 15).

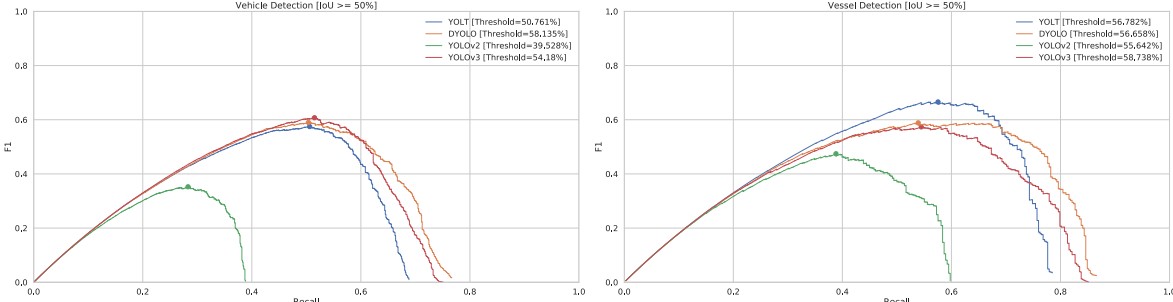

**Figure 13.** F1-curves of our detectors.

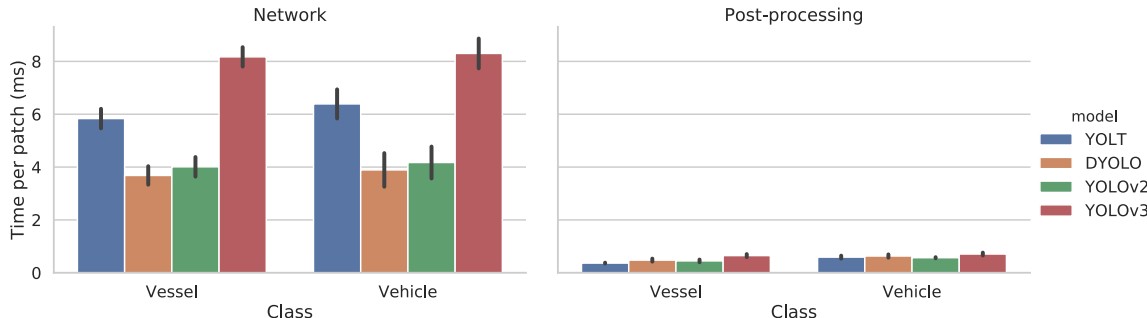

**Figure 14.** Inference timing results of our models with their threshold set at the best F1 value, averaged for a 416 × 416 pixel patch. This test was measured on the entire test dataset with an Nvidia GTX 1080 Ti GPU.

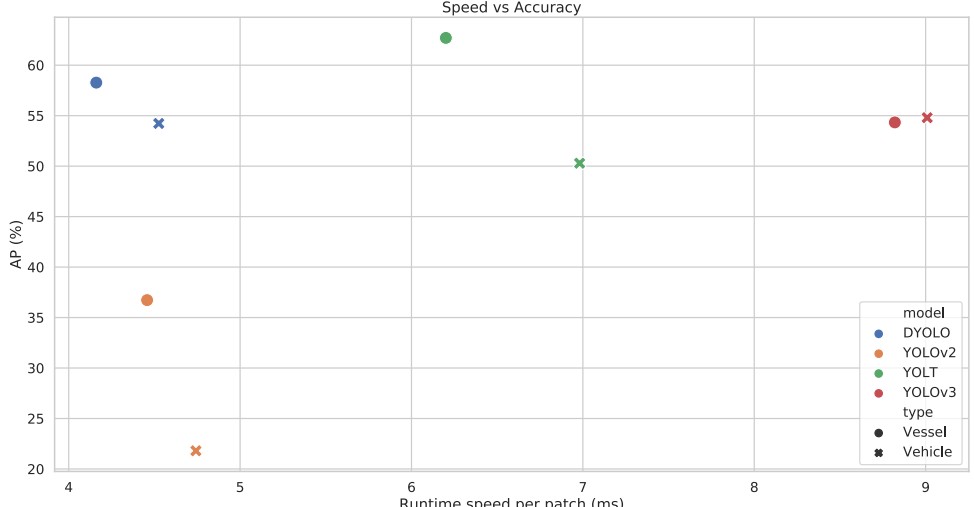

**Figure 15.** Speed vs. accuracy on the test set of our data. Accuracy is measured from the models with pretrained weights from ImageNet.

## 4. Discussion

In this study, we performed a practical implementation and experimental comparison on detection and classification of vehicles and vessels using optical satellite imagery with spatial resolutions of 0.3–0.5 m. A series of experiments was performed to understand the advantages and weaknesses of each one of the tested models.

In general, terms, YOLOV2 appears to be the less accurate to deal with the detection of small objects, while the three other networks seem to reach more similar performance (see Table 7). One remarkable finding is that for the specific case of vessel detection, YOLT outperforms both YOLOV3 and D-YOLO by a significant margin; however for vehicle detection it is the exact opposite. When looking at the accuracy of the models with pretrained DOTA weights, we can see that this difference has disappeared. This can be explained by looking at the different network architectures. The YOLT network is smaller and thus needs less data to properly train, while D-YOLO is deeper. The results from DOTA do show that with more data, D-YOLO seems to be a more capable network, reaching significantly better results than YOLT for both cases of vehicle and vessel detection. The dataset ablation study seems to support this hypothesis, as more data generally leads to better results in that experiment as well. In the future it might be interesting to also train YOLOV3 on the DOTA dataset and use those weights, as YOLOV3 is an even deeper network which might benefit from more data as well. Another alternative is to simply annotate more satellite images, creating a bigger dataset, thus eliminating the need for pretrained DOTA weights, which cannot be used outside of academic research.

**Table 7.** Overview of the speed and accuracy of the different models, measured on the entire test set. Speed is measured per 416 × 416-pixel patch.

|  | **Vehicles** | | | | **Vessels** | | | |
|---|---|---|---|---|---|---|---|---|
|  | **YOLT** | **D-YOLO** | **YOLOV2** | **YOLOV3** | **YOLT** | **D-YOLO** | **YOLOV2** | **YOLOV3** |
| $AP_{ImageNet}$ (%) | 50.29 | 54.23 | 21.80 | 54.80 | 62.70 | 58.27 | 36.73 | 54.33 |
| $AP_{DOTA}$ (%) | 52.22 | 59.52 | 22.41 | - | 62.02 | 66.02 | 46.57 | - |
| Speed per patch (ms) | 6.98 | 4.53 | 4.74 | 9.01 | 6.20 | 4.16 | 4.46 | 8.82 |

When we observe that both YOLOV3 and D-YOLO achieve a similar performance, the deciding factor might be inference time. When comparing runtime speeds, D-YOLO outperforms YOLOV3

by a factor of two and is thus a better fit for the task of satellite-based vehicle and vessel detection. This counters the general belief that deeper networks are better, as D-YOLO has significantly less convolutional layers and still reaches the same or a better accuracy in our experiments.

One might wonder whether we trained the networks properly and although we did fine-tune the hyperparameters to the best of our ability, the fact that we tuned those hyperparameters on the GE subset of our data, might not give the best results overall. We do believe this decision to be justified as it introduces less bias in the experiments, but investigating the influence of tweaking on a subset of data is something that needs more research in the future. Performing a full exhaustive search on the hyperparameters—while taking a long time—might also prove to be beneficial to reach even better accuracy, but care must be taken not to overfit the test set, which might result in a lower real accuracy.

For the case of multi-label detection, we conclude that this task is rather challenging for the detectors examined in this study. It is noted, however, that the classification scheme used here is very demanding, as regards the size of the minimum detectable object with respect to the spatial resolution of the dataset. In addition, there is a class imbalance, which makes training on specific classes very difficult, therefore a bigger dataset might be necessary to create useful results. Another issue to consider is annotator bias. This dataset was created by a single person and is thus possibly biased towards that person's background (*i.e.* computer vision engineer). While this problem is not so bad for the detection part, as vehicles and vessels are usually clearly distinguishable in this data, correctly classifying these objects has proven to be really challenging. Besides increasing the size of this dataset, it might be necessary to have these objects labeled by multiple experienced image analysts and to average the different results together. One final note about this is that while deep learning can achieve impressive results and even outperform humans at specific tasks, it cannot detect things that are completely undetectable by humans in any way. Indeed, a lot of satellite analysts use external information, like port locations or even GPS data of vessels, to label them. While this might be nice to speed up the annotation process and generate objectively better labels, there must still be some kind of visual clue the detector can exploit, to correctly classify these objects. Another approach might be to perform some kind of data fusion, combining e.g., GPS localization data with our image data, in order to create a better classification model.

## 5. Conclusions

The main contributions of this work are that we demonstrated the feasibility of vehicle and vessel detection in optical satellite imagery with a variable spatial resolution between 0.3 and 0.5 m, in which these objects are down to a few pixels in size. We selected four single-shot detection network architectures for their fast execution time and compared these, in the meantime optimizing the many hyperparameters for such a case of small object detection in large images. We empirically showed that there is a need to tune the hyperparameters differently for satellite object detection, to reach a good accuracy. A good understanding of the different hyperparameters is primordial for such a task, and thus we explained some of the hyperparameters specific to the loss function of single-shot detection networks. The implementation of these four models can be found in our open-source Lightnet library [22].

From our results, we can conclude that D-YOLO seems to be the most optimal detector, reaching the highest accuracy ($AP_{vehicle}$: 60%, $AP_{vessel}$: 66%) and fastest runtime speeds ($\pm 4$ ms per $416 \times 416$ patch). While our best results might not be good enough for fully automated detection pipelines, they can already be deployed as a tool for helping data analysts, speeding up their workflow tremendously. It is in this setting that the speed of our networks is primordial, as it promotes a convenient and fast workflow for data analysts and allows them to keep on working with the tool without long waiting times.

The problem of automatic satellite object detection can certainly not be considered solved. There is a dire need for bigger datasets with lots of variability, and even more so for the step of fine-grained classification, where we also need a more balanced dataset, which has multiple examples of each class

of vehicle and vessel. The aforementioned technique of using already existing object detection models, such as the ones presented in this paper, but keeping a human in the loop as supervisor could prove to be a valuable tool in order to more easily scale up datasets, after which we might be able to create stronger models, capable of fully automatic small object detection in satellite imagery.

**Author Contributions:** Conceptualization, S.P., V.K. and J.-P.R.; Data curation, T.O., S.P., V.K. and J.-P.R.; Formal analysis, T.O.; Funding acquisition, S.P. and T.G.; Investigation, T.O.; Methodology, T.O. and S.P.; Project administration, S.P., V.K. and T.G.; Software, T.O.; Supervision, S.P. and T.G.; Validation, T.O., S.P., V.K. and J.-P.R.; Visualization, T.O.; Writing—original draft, T.O.; Writing—review and editing, S.P., V.K. and T.G. All authors have read and agreed to the published version of the manuscript.

**Funding:** This work was partially funded by the EU (Tender Lot 6 SatCen RP-01-2018) and FWO (SBO Project Omnidrone).

**Acknowledgments:** We thank the SADL research group of KU Leuven for their project managing efforts during this project and help with annotating.

**Conflicts of Interest:** The authors declare no conflict of interest. The funders provided the raw image data, outlined their needs and current workflow to assess the necessary research and, finally, helped in the writing of this manuscript.

**Disclaimer:** The views and opinions expressed in this article are those of the authors solely and do not reflect any official policy or position of their employers.

## Abbreviations

The following abbreviations are used in this manuscript:

| | |
|---|---|
| AI | Artificial Intelligence |
| AIS | Automatic Identification System |
| AP | Average Precision |
| CNN | Convolutional Neural Network |
| COTS | Commercial Off-The-Shelf components |
| D-YOLO | Deconvolutional YOLO; A variation of the YOLOV2 detector |
| DOTA | Dataset for Object Detection in Aerial images |
| GPS | Global Positioning System |
| GPU | Graphics Processing Unit |
| IMINT | Image Intelligence |
| IoU | Intersection over Union |
| PR | Precision-Recall curve |
| RGB | Red–Green–Blue; channels of an image |
| R-CNN | Regions with CNN features; An example of a two-staged detector |
| YOLO | You Only Look Once; An example of a single-shot detector |
| YOLT | You Only Look Twice; A variation of the YOLOV2 detector |

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
