# Peer review of "Vehicle and Vessel Detection on Satellite Imagery: A Comparative Study on Single-Shot Detectors"

_remotesensing, doi:10.3390/rs12071217_

Round 1

Reviewer 1 Report

  • a space is needed between values and units
  • first person should not be used in the papers
  • sub sections to follow should be introduced. I.E. two headings should not be next to each other. E.g. between heading 2 and 2.1 should introduce sub-sections to follow. This must be done for all similar cases.
  • "500x500" would read better as 500 x 500, and for similar other cases
  • similarly, "0.3-0.5m" would read better as "0.3 m - 0.5 m", and for similar cases in this paper.
  • It will be useful to have in the discussion/conclusion a table that summerizes the results of the different models investigated and compared. This table and comparison can then be discussed and allow the reader to see a full comparison and overview.
  • The contributions should be re-iterated in the conclusion, and eleborated how they were achieved.

Author Response

Dear Reviewer,

Please see the attachment below for our response to your review.

Kind regards

Reviewer 2 Report

1. The title, this work is just a Comparative work, without any creative work? 2. IN the abstract, “ with objects ranging 4 from 5 to a few 100 pixels in length”. The pixels are length or area? 3. IN Section 3, How many satellite images have you used? Only the two in Fig.6. But I think the images in Fig.6 are not the entire scene. Maybe the rest scene has the background which is prone to false alarms. So the other scene should be detected, to verify the real ability for satellite images. 4. I think the comparison experiments on time consumption should also be performed. I am very interesting about how long time to process a satellite image? 5. And before detection, is there any preprocessing operations which is not included, like sliding? 6. I'm not sure if an article just compares the performance of four methods is enough. (Actually only one method YOLO, the other there can be seen as its variants). If the experiment part cannot be more sufficient and reasonable, I think the content of this manuscript is not enough.

Author Response

(The authors gave the same response as above.)

Reviewer 3 Report

Review of article:  RemoteSensing-752511

Article Title:  Vehicle and Vessel Detection on Satellite Imagery: a Comparative Study on Deep Learning Algorithms

This paper investigates how four (existing) different single-shot object detectors (YoloV2, YoloV3, D-Yolo and Yolt) handle the task of vehicle and vessel detection and classification in satellite imagery with a spatial resolution between 0.3 and 0.5m, where objects in an image appear only a few pixels large.  

COMMENTS

The paper is interesting well written and presented. The authors have done a good comparative study of the single-shot detectors: YoloV2, YoloV3, D-Yolo and Yolt.

In my view the material on two-stage and single-shot detectors, appearing in the introduction as a number of citations, needs more elaboration. A 2-4 lines short presentation of the work done in each of the cited publications is needed. Just citing a number of references is not good enough.

I would like the authors to shortly explain how exactly they located the values of hyper-parameters. By trial and error? Exhaustive experimentation? Can you demonstrate that the selected values are optimal?

The authors systematically refer to the four compared models as “our models” throughout the entire paper. This is misleading, please use “the four models”, “the compared models” or something which does not imply that they are “your models”.

Typo: page 17, line 326 “… the needs …” à “… that needs …”

Author Response

(The authors gave the same response as above.)

Round 2

Reviewer 2 Report

I think the authors have solved most of my concerns, but for Question 3, I still think the result can be improved:

Of course the entire image is too big for readers to see the objects. But you can show the entire image with low resolution, and the subimages can show some enlarged local areas, for example:

1. local area with objects of interest;

2. local area which id prone to false alarm

Author Response

Dear reviewer,   We would like to thank you again for the time you spent reviewing our paper and for the constructive feedback you provided. We refactored our paper, with regards to your comment.  

I think the authors have solved most of my concerns, but for Question 3, I still think the result can be improved:

Of course the entire image is too big for readers to see the objects. But you can show the entire image with low resolution, and the subimages can show some enlarged local areas, for example:

  • local area with objects of interest
  • local area which id prone to false alarm
  We are not allowed to show the entire image as this is private data, but we adapted Figure 1 to show a more zoomed out view, and provide subimages to show some enlarged local areas. We think that this indeed more clearly communicates the kind of data we are working with and the challenges that it brings.   Kind Regards